# Impact of Negative Symptoms on Functioning and Quality of Life in First Psychotic Episodes of Schizophrenia

**DOI:** 10.3390/jcm11040983

**Published:** 2022-02-14

**Authors:** Lorena García-Fernández, Verónica Romero-Ferreiro, Luis Sánchez-Pastor, Mónica Dompablo, Isabel Martínez-Gras, Juan Manuel Espejo-Saavedra, David Rentero, Ana Isabel Aparicio, Miguel Angel Alvarez-Mon, Guillermo Lahera, Jimmy Lee, Jose Luis Santos, Roberto Rodriguez-Jimenez

**Affiliations:** 1Clinical Medicine Department, Universidad Miguel Hernández, 03550 Alicante, Spain; lorena.garciaf@umh.es; 2Psychiatry Department, Hospital Universitario de San Juan, 03550 Alicante, Spain; 3Biomedical Research Networking Centre in Mental Health (CIBERSAM), 28029 Madrid, Spain; mvromero@ucm.es (V.R.-F.); monicadompablo@gmail.com (M.D.); davidrente7@hotmail.com (D.R.); aiaparicio@sescam.jccm.es (A.I.A.); guillermo.lahera@gmail.com (G.L.); joseluis.santosg@gmail.com (J.L.S.); 4Quality and Academic Compliance Unit, Universidad Europea de Madrid, 28670 Madrid, Spain; 5Instituto de Investigación Sanitaria Hospital 12 de Octubre (Imas 12), 28041 Madrid, Spain; lspastor@salud.madrid.org (L.S.-P.); isabelmgras@gmail.com (I.M.-G.); juanmaespejosaavedra@gmail.com (J.M.E.-S.); 6Cardenal Cisneros, Centro de Enseñanza Superior Adscrito a la Universidad Complutense de Madrid, 28040 Madrid, Spain; 7RETIC (Network of Addictive Conditions), Institute of Health Carlos III, 28029 Madrid, Spain; 8Legal Medicine, Psychiatry and Pathology Department, Universidad Complutense de Madrid, 28040 Madrid, Spain; 9Psychiatry Department, Hospital Virgen de la Luz, 16002 Cuenca, Spain; 10Neurobiological Research Group, Institute of Technology, Universidad de Castilla-La Mancha, 16071 Cuenca, Spain; 11Department of Medicine and Medical Specialities, Faculty of Medicine and Health Sciences, University of Alcalá, 28801 Alcalá de Henares, Spain; maalvarezdemon@icloud.com; 12Department of Psychiatry and Mental Health, Hospital Universitario Infanta Leonor, 28031 Madrid, Spain; 13Research Division, Institute of Mental Health, Singapore 539747, Singapore; jimmy_lee@imh.com.sg; 14North Region & Department of Psychosis, Institute of Mental Health, Singapore 539747, Singapore; 15Lee Kong Chian School of Medicine, Nanyang Technological University, Singapore 639798, Singapore

**Keywords:** negative symptoms, expressive deficits, experiential deficits, functioning, quality of life, first psychotic episode

## Abstract

Negative symptoms are not considered a unitary construct encompassing two different domains, diminished expression, and avolition-apathy. The aim of this study was to explore the relationships between each domain and psychosocial functioning and quality of life in people with a first psychotic episode of schizophrenia. In total, 61 outpatients were assessed with the Clinical Assessment Interview for Negative Symptoms (CAINS), The Functioning Assesment Short Test (FAST) and The Quality of Life Scale (QLS). The mean global score for CAINS was 21.5 (SD: 15.6), with a CAINS Avolition-Apathy (MAP) score of 17.0 (SD: 11.8), and CAINS Diminished Expression (EXP) score of 4.5 (SD: 5.0). The mean FAST score was 31.9 (SD: 18.9), and 41.1 (SD: 17.9) for QLS. Linear regression analysis revealed a significant (F(4,53) = 15.65, *p* < 0.001) relationship between MAP and EXP CAINS’ score and FAST score. CAINS-MAP was more predictive of FAST scores (β = 0.44, *p* = 0.001) than CAINS-EXP (β = 0.37, *p* = 0.007). Linear regression analysis for QLS revealed a significant model (F(4,56) = 29.29, *p* < 0.001). The standardized regression weight for the CAINS-MAP was around three times greater (β = −0.63, *p* < 0.001) than for CAINS-EXP (β = −0.24, *p* = 0.024). The two different domains are associated differently with functionality and quality of life.

## 1. Introduction

Negative symptoms of schizophrenia constitute a therapeutic challenge, as well as one of the main areas to consider in order to improve functioning (proper behaviors in real-world social situations) [1] and quality of life (the individual’s perception of their position in life in the context of the culture and value systems) [2] in people with their first psychotic episodes of schizophrenia [3]. Despite the improvements achieved in the care of people with schizophrenia, up to one-third of patients might have idiopathic and stable negative symptoms [4]. From the first description of the disorder recorded by Morel in 1852 [5], negative symptoms have long been recognized as a core and clinically meaningful feature of schizophrenia. Deficit symptoms were first proposed by Kraepelin and Bleuler, referring to basic symptoms of affective blunting and weakening of emotional activities [6,7]. The subsequent identification, under a dichotomous perspective of negative symptoms as opposed to the positive ones, was defined by Crow in the 1980s including blunted affect and poverty of speech, and later revised by Andreasen with the incorporation of avolition, anhedonia, asociability and attentional deficit, suggesting the existence of a different clinical phenotype and pathophysiological substrate when negative symptoms were predominant [8,9]. Shortly after, Carpenter advanced the concept of deficit schizophrenia and, differentiated between primary and secondary negative symptoms depending on whether they were inherent to the disease or the result of additional factors such as emotional reactions, mood disorders, pharmacological treatment or the response to environmental events. The presence of at least two of the following was further suggested: restricted affect, diminished emotional range, poverty of speech, curbing of interest, diminished sense of purpose and diminished social drive, as part of the diagnostic criteria for the deficit syndrome [10,11].

With this perspective in mind, the National Institute of Mental Health (NIMH) organized in 2005, the Consensus Development Conference on Negative Symptoms with the aim of favoring the development of evidence-based measures and treatments for negative symptoms [12]. Results from the international discussion identified affective flattening, alogia, asociality, avolition and anhedonia as domains of negative symptoms and achieved a clearer definition of a hierarchical 5-factor model consistent with the NIMH-MATRICS domains with expressive and experiential domains, as second-order factors [13]. In addition, they fostered the development of new assessment instruments to address the limitations of the existing tools and encouraged the development of different therapeutic targets. As a consequence, two next-generation negative symptom scales that include an adequate sampling of expressive and experiential deficits domains were developed: The Brief Negative Symptom Scale (BNSS) [14,15] and the Clinical Assessment Interview for Negative Symptoms (CAINS) [16,17].

Negative symptoms represent a loss of normal brain functioning and have long been recognized as the most devasting among all clinical features in schizophrenia [18]. However, they are not considered a unitary construct encompassing two different domains, diminished expression including alogia and blunted affect, and avolition-apathy referring to experiential deficits, including asociality, avolition and anhedonia [19,20,21], that have been consistently replicated regardless of the measurement instrument used [22,23,24]. Negative symptoms are present in the prodromal and initial stages of the disease, are persistent, worsen with age [9], behave independently of cognition or affectivity [25], present a lack of response to pharmacological treatment [26], and importantly, are the most distressing for the family and the main determinants of impairment in functioning and quality of life, which have now become the main therapeutic targets in people with schizophrenia [27,28,29] both in early and chronic stages [8,27,29,30,31,32].

We hypothesize that the strength of the association between the two different domains of negative symptoms and both functionality and quality of life might be different. Unfortunately, there is a paucity of studies exploring the relationship that the expressive and experimental domains of negative symptoms exert on psychosocial functioning and quality of life in people with a first psychotic episode of schizophrenia using the Clinical Assessment Interview for Negative Symptoms (CAINS). Given this interest, the aim of the present study was to explore the strength of the relationships between each domain of negative symptoms as separate constructs and psychosocial functioning and quality of life in people with a first psychotic episode of schizophrenia.

## 2. Materials and Methods

### 2.1. Sample

The present cross-sectional study was carried out with the participation of 61 Caucasian outpatients with a first psychotic episode of schizophrenia, who were consecutively included in the First Episode Programs of the Universitary “12 de Octubre” Hospital (Madrid, Spain) and “Virgen de la Luz” Hospital (Cuenca, Spain). The inclusion criteria included: (1) diagnosis of schizophrenia or schizophreniform disorder according to DSM-5 criteria [33], using the Structured Clinical Interview for DSM-5 (SCID-5) [34], (2) a minimum of eight consecutive weeks of stabilization on their antipsychotic medication after discharge from the hospitalization unit, (3) aged from 18 to 55 years and (4) fluent Spanish speaking that enables the protocol to be completed. Exclusion criteria were: (1) substance use disorder diagnosed in the past eight weeks (excluding nicotine and caffeine), and (2) traumatic head injury. All participants were clinically assessed by psychiatrists with more than 5 years of experience in the use of the scales. The study was approved by the Clinical Research Ethics Committee and all participants signed the informed consent. Demographic and clinical characteristics of participants are described in Table 1.

### 2.2. Assessment Instruments

#### 2.2.1. Symptoms Were Assessed Using the Positive and Negative Syndrome Scale

(PANSS) [35] applied only for descriptive purposes.

#### 2.2.2. The Clinical Assessment Interview for Negative Symptoms (CAINS) 

The Clinical Assessment Interview for Negative Symptoms (CAINS) is a 13-item tool designed to address the limitations inherent to previous assessment instruments used to evaluate negative symptoms [36,37]. The scale provides both a single summary score, and two scores for the two negative symptom domains [17] reporting the emotional experience (motivation and pleasure) and emotional expression subscales separately. The first nine items, the motivation and pleasure (MAP) subscale, relate to experiential deficits, assessing the motivation, anticipation and experience of pleasure in occupational and recreational activities, and social contacts with partners, friends and family. The last four items, the expression (EXP) subscale, relate to expressive deficits, assessing both vocal and gestural features. All items are rated on a scale of 0–4, with higher scores reflecting greater impairment. 

#### 2.2.3. The Functioning Assessment Short Test (FAST)

The Functioning Assessment Short Test (FAST) is a 24-item instrument divided into 6 specific areas of operation (autonomy, occupational functioning, cognitive functioning, financial aspects, relationships and free time) designed to address functioning in patients with mental disorders [38,39,40]. All items are rated on a scale of 0–3, with higher scores reflecting greater operating difficulties.

#### 2.2.4. The Quality of Life Scale (QLS)

The Quality of Life Scale (QLS) is a 21-item semi-structured interview designed to measure quality of life specifically in patients with schizophrenia [41,42,43]. The scale obtains information about symptoms and functioning in relation to four areas: interpersonal relationships, instrumental role, intrapsychic functions, and use of common objects and daily activities. It provides an overall score as well as single scores on each of the 4 factors. Each item is scored from 0 (greater degree of dysfunction) to 6 (normality). The higher the score, the better the patient’s functioning in that category. 

### 2.3. Statistical Analysis

Data were managed and analyzed with SPSS v.24. Mean and SD were used for continuous variables and percentages for categorical variables. Multiple regression analysis (ENTER method) was employed to develop a predictive model of FAST total scores for the MAP and EXP subscales of the CAINS, included as predictor variables. Second, another regression model was performed, this time considering QLS scores as the response variable. Age and gender (0 = female, 1 = male) were included as covariates in the regression models. Collinearity diagnostics were based on the variance inflation factor (VIF). The absence of collinearity was considered when VIF values were lower than 4 [44].

## 3. Results

### 3.1. Relationship between Negative Symptoms and Functioning

Linear regression analysis revealed a statistically significant (F(4,53) = 15.65, *p*< 0.001) relationship between CAINS’ MAP and EXP subscales and FAST scores. The proportion of variance in FAST scores explained by the model was substantial (adjusted r^2^ = 0.507). None of the two independent variables were correlated, as the variance inflation factor is closer to 1 (VIF = 1.84 and 1.99, respectively). Based on the results of the above model, the formula to calculate the predicted FAST score is: Predicted FAST = [34.15 + 0.72 CAINS-MAP + 1.40 CAINS-EXP − 6.14 Gender − 0.62 Age].

From the standardized regression weights (β) in the model, the CAINS-MAP had a slightly greater effect on FAST scores (β = 0.44, *p* = 0.001) than the CAINS-EXP (β = 0.37, *p* = 0.007).

### 3.2. Relationship between Negative Symptoms and Quality of Life

A linear regression analysis using the QLS overall score was performed. Results revealed a statistically significant model (F(4,56) = 29.29, *p* < 0.001) between the CAINS’ MAP and EXP subscales and QLS scores. The proportion of variance explained by the model was large (adjusted r^2^ = 0.654). Regarding collinearity diagnostic, none of the two independent variables proved to be correlated, as the variance inflation factor is, again, closer to 1 (VIF = 1.82 and 1.89, respectively). Based on the results of the above model, the formula to calculate the predicted QLS score is: Predicted QLS = [80.65 − 1.36 CAINS-MAP − 1.21 CAINS-EXP − 3.13 Gender − 0.56 Age].

Negative weights indicate that higher scores on quality of life are associated with lower CAINS scores. From the standardized regression weights, CAINS-MAP had a larger effect on QLS (β= −0.63, *p* < 0.001) than CAINS-EXP (β = −0.24, *p* = 0.024).

Table 2 shows the relationship between negative symptoms and functioning and quality of life.

## 4. Discussion

This study highlights the need to put higher emphasis on understanding the structure of negative symptoms and its influence on the psychosocial functioning and quality of life of people with a first psychotic episode of schizophrenia, which is essential to improve their future.

The aim of the present study was to explore the strength of the relationship between each domain of negative symptoms as separate constructs and psychosocial functioning and quality of life in people with a first psychotic episode of schizophrenia. Our results have shown that both MAP and EXP subscales, explored through the CAINS, are associated with functioning and quality of life.

Regarding psychosocial functioning, both domains of negative symptoms are independently related to functional performance with a slightly greater predictive weight for the MAP subscale, suggesting that it may represent a more severe aspect of psychopathology. In line with this, findings from chronic schizophrenia, first episode psychosis and clinical high risk for psychosis have also found experiential deficits to be linked to various aspects of functioning, both cross-sectionally [3,45,46,47,48,49] and longitudinally [50,51,52,53,54]. 

Similar to functioning, our study has evaluated the different negative symptoms’ domains, both expressive and experimental, and correlated them with quality of life. We found that both the MAP and EXP subscales were independently associated with quality of life. Moreover, the impact of MAP’s score on QLS compared with the EXP score was almost triple in people with a first psychotic episode, which makes this domain of negative symptoms a priority intervention target to improve quality of life in early stages and also in chronic schizophrenia [55].

In line with the obtained results, it could be proposed that the two symptomatic domains of negative symptoms explore different psychopathological areas. On the one hand, the EXP subscale is related to the observation of emotional expression, whereas the MAP subscale is related to more internal aspects of the emotional experience such as lack of will, lack of pleasure and absence of motivational goals that will further limit successful interaction between people with a first psychotic episode and society.

Globally, the pattern of findings across relationships between negative symptoms’ domains and both functioning and quality of life represents a distinct and greater predictive power for the MAP subscale compared with the EXP subscale, which gives the experential deficits domain higher impact on severity and greater weight in outcome, enriching previous research showing that those patients with a predominant MAP subscale score had, in addition, significantly more severe conceptual disorganization, greater social cognition impairment, higher rates of hospitalization and poorer social functioning [56]. These data bring consistency to previous findings [57,58,59,60] and provide the novelty of showing a different link for each domain within the construct of negative symptoms with both functional outcomes and quality of life in people with a first psychotic episode of schizophrenia. However, the specific weight of each domain, a novelty provided by the study presented, has not been evaluated separately.

Moreover, our results give support to DSM 5 [33,61] positioning diminished expression and avolition, anhedonia and asociality as the two prominent domains of negative symptoms [21], given their importance in the prodromal and residual phases and the huge burden they impose on functioning and quality of life across the life course of people with schizophrenia.

The main strengths of the present study are: first, the use of a rigorous negative symptom assessment instrument that systematically measures experiential and expressive symptoms following DSM 5′s 2 factors model; second, the evaluation of people in early stages of psychosis avoiding bias due to chronicity; and finally, the use of two main clinically variables, functioning and quality of life, as outcome. Despite the above, some limitations should be mentioned: first, the cross-sectional design preclude determination of direct causal relationships; second, a larger sample of participants could have provided more robust conclusions; and finally, participants’ cognitive performance might have been impacting on the functioning and quality of life. 

Advances in the understanding of negative symptoms have led to the identification of two interrelated yet separable domains, diminished expression and experiential deficits, both in patients’ first psychotic episodes [58] and chronic schizophrenia [19,59] that might necessitate different therapeutic approaches [19,20,62,63] emphasizing the benefit of measuring them separately [16]. We believe that more emphasis should be placed on rigorously assessing negative symptoms and the development of specific treatments for each domain as part of the clinical protocols for patients with a first psychotic episode in order to improve their functioning and quality of life.

## Figures and Tables

**Table 1 jcm-11-00983-t001:** Demographic and clinical characteristics of participants with a first psychotic episode of schizophrenia.

	Patients (*n* = 61)
Age years mean (SD)	26.5 (8.2)
Gender *n* (% men)	44 (72.1)
Education years mean (SD)	12.3 (3.0)
Second Generation Antipsychotics *n* (%)	59 (96.7%)
Clorpromazine equivalents-mg mean (SD)	413.2 (238.0)
PANSS	
Positive mean (SD)	11.2 (4.8)
Negative mean (SD)	16.8 (7.3)
General Psychopathology mean (SD)	28.0 (8.5)
Total mean (SD)	56.0 (17.2)
CAINS	
Avolition-apathy mean (SD)	17.0 (11.8)
Diminish expression mean (SD)	4.5 (5.0)
Total mean (SD)	21.5 (15.6)
FAST mean (SD)	31.9 (18.9)
QLS mean (SD)	41.1 (17.9)

PANSS: The Positive and Negative Syndrome Scale. CAINS: The Clinical Assessment Interview for Negative Symptoms. FAST: The Functioning Assessment Short Test. QLS: The Quality of Life Scale.

**Table 2 jcm-11-00983-t002:** Negative symptoms as predictive variables for functioning and quality of life.

Response Variable	Parameter	Unstandardized Coefficient (Std. Error)	Beta (Standardized)	*p*-Value
FAST r^2^ = 0.507	Intercept	34.15 (7.23)		<0.001
	CAINS-MAP	0.72 (0.20)	0.443	0.001
	CAINS-EXP	1.40 (0.50)	0.37	0.007
	Sex	−6.14 (4.29)	−0.143	0.159
	Age (years)	−0.62 (0.22)	−0.274	0.006
QLS r^2^ = 0.654	Intercept	80.65 (7.54)		<0.001
	CAINS-MAP	−1.36 (0.22)	−0.634	<0.001
	CAINS-EXP	−1.21 (0.52)	−0.242	0.024
	Sex	−3.13 (4.43)	−0.056	0.484
	Age (years)	0.56 (0.24)	0.181	0.023

CAINS-MAP: Avolition-apathy domain. CAINS-EXP: Diminish expression domain. FAST: The Functioning Assessment Short Test. QLS: The Quality of Life Scale.

## Data Availability

The data from this study will be made publicly available following the completion of all publications and following the removal of all identifiers by request.

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
