# Peer review of "Impact of Negative Symptoms on Functioning and Quality of Life in First Psychotic Episodes of Schizophrenia"

_jcm, 2022, doi:10.3390/jcm11040983_

Round 1

Reviewer 1 Report

Thank you very much for inviting me to review this paper examining the impact of negative symptoms on functioning and quality of life in patients with first-episode schizophrenia. The study design and statistical analyses are adequate, and the findings contribute to advancing knowledge in this field. I recommend publication after the following concerns have been successfully addressed:

1) It is important to indicate in the Introduction that the prevalence of primary and enduring negative symptoms (deficit schizophrenia) is around 33% in schizophrenia cohorts. Cite: PMID: 29422932.

2) I is also important to mention that the two-dimensional model of negative symptoms (avolition/apathy and expressive deficit) have been consistently replicated regardless of the measurement instrument used to evaluate negative symptoms. Cite: PMID: 10080556; PMID: 23472837; PMID: 32149549; PMID: 30791343.

3) Positive symptoms and general psychopathology (according to the PANSS) should be included as covariates in the regression models (using the ENTER method). If for any reason the authors could not perform these analyses, they should comment on it in the limitations.

4) In limitations, the authors should add that the inherent limitations of a cross-sectional design preclude determination of direct causal relationships. Another limitation that should be mentioned is that no neurocognitive evaluation was made of the patients (it is well known that cognitive deficits have a negative impact on functioning and quality of life of patients with psychosis).

5) The sample size is small (n = 61) considering that the study was done in two hospitals. This may be due to a short recruitment period and/or inclusion/exclusion criteria (patients with first-episode of schizophrenia and without substance use). The authors should mention something in this respect to clarify it.

Author Response

Reviewer 1

Comments and Suggestions for Authors

Thank you very much for inviting me to review this paper examining the impact of negative symptoms on functioning and quality of life in patients with first-episode schizophrenia. The study design and statistical analyses are adequate, and the findings contribute to advancing knowledge in this field. I recommend publication after the following concerns have been successfully addressed:

1) It is important to indicate in the Introduction that the prevalence of primary and enduring negative symptoms (deficit schizophrenia) is around 33% in schizophrenia cohorts. Cite: PMID: 29422932.

Thank you for the recommendation, we have reflected it in the introduction section. Without a doubt, the data provides valuable information (page 1 and 2; lines 46-48)

2) I is also important to mention that the two-dimensional model of negative symptoms (avolition/apathy and expressive deficit) have been consistently replicated regardless of the measurement instrument used to evaluate negative symptoms. Cite: PMID: 10080556; PMID: 23472837; PMID: 32149549; PMID: 30791343.

 Thank you for your suggestion, the information has been incorporated in the manuscript and references have been also included, (page2; lines 82-83).

3) Positive symptoms and general psychopathology (according to the PANSS) should be included as covariates in the regression models (using the ENTER method). If for any reason the authors could not perform these analyses, they should comment on it in the limitations.

During the initial approach, several covariates such as clinical (positive symptoms, general psychopathology, milliequivalents of chlorpromazine, premorbid functioning), and sociodemographic (for example, employment, marital status) variables were considered. Finally, and based on the sample size, we decided to include just gender and age, leaving the rest of the variables for an ongoing study with a higher "n" (which allows to include more covariates in the regression). Despite this, we re-analyzed our data including -as Reviewer 1 suggests- PANSS-P and PANSS-GP scores. However, given the high correlation between both scales (r = 0.74, p < 0.001) the variance inflation factor (VIF) increases reaching values over the established limit of 4. For that reason, we have chosen not to include these covariates.

4) In limitations, the authors should add that the inherent limitations of a cross-sectional design preclude determination of direct causal relationships. Another limitation that should be mentioned is that no neurocognitive evaluation was made of the patients (it is well known that cognitive deficits have a negative impact on functioning and quality of life of patients with psychosis).

 Thanks for the suggestion, limitations noted by Reviewer 1 have been incorporated as study limitations (page 6; lines 232-235).

5) The sample size is small (n = 61) considering that the study was done in two hospitals. This may be due to a short recruitment period and/or inclusion/exclusion criteria (patients with first-episode of schizophrenia and without substance use). The authors should mention something in this respect to clarify it.

Reviewer 1is right, a larger sample size would have been desirable.

Substance use disorder has been specifically defined and included in the methods sections (page 3; line 109).

Moreover, the small sample size has been reported as a limitation, (page 6; line 232).

Reviewer 2 Report

This is an interesting study detailing the strength of association between various aspects of negative symptoms in first episode psychosis and functioning/quality of life. This topic is significant as negative symptoms typically are persistent and burdensome, and improvements to quality of life/functioning are valued by patients as much (or even more than) symptom reduction. The statistical methods are appropriate.

Please provide a definition of quality of life as compared to functioning, as these concepts share some conceptual overlap in the introduction, and provide justification for the hypothesis that the two aspects of negative symptoms will share different relationships with these constructs.

Can you please clarify in the methods section whether participants were recruited following their first episode of psychosis? I am aware that they were recruited from this program, but in some instances this service encompasses people early in the trajectory of illness (not just FEP specifically).

Can authors clarify why any substance use (and not just substance use disorder) was an exclusion criteria? No alcohol in the prior 8 weeks seems highly restrictive and likely to result in a biased sample.

Discussion would be enriched by a deeper exploration of why MAP versus EXP domains would be differentially related to quality of life and functioning.

Minor comments

Line 67 – should be “achieved”

Author Response

Reviewer 2

Comments and Suggestions for Authors

This is an interesting study detailing the strength of association between various aspects of negative symptoms in first episode psychosis and functioning/quality of life. This topic is significant as negative symptoms typically are persistent and burdensome, and improvements to quality of life/functioning are valued by patients as much (or even more than) symptom reduction. The statistical methods are appropriate.

Please provide a definition of quality of life as compared to functioning, as these concepts share some conceptual overlap in the introduction, and provide justification for the hypothesis that the two aspects of negative symptoms will share different relationships with these constructs.

Following the request of Reviewer 2, the definitions of functioning and quality of life have been incorporated to the manuscript (page 1; lines 45-47).

 Can you please clarify in the methods section whether participants were recruited following their first episode of psychosis? I am aware that they were recruited from this program, but in some instances this service encompasses people early in the trajectory of illness (not just FEP specifically).

Indeed, only patients who meet criteria for schizophrenia and/or schizophrenia have been included . On the other hand, high-risk mental states have not been incorporated into this study. This has been reflected in the methodology section as follows:

“diagnosis of schizophrenia or schizophreniform disorder according to DSM-5 criteria [27], using the Structured Clinical Interview for DSM-5” (page 3; line 107-108)

Can authors clarify why any substance use (and not just substance use disorder) was an exclusion criteria? No alcohol in the prior 8 weeks seems highly restrictive and likely to result in a biased sample.

 Reviewer 2 is right, thanks for the input. Specifically, people with a substance use disorder diagnosed in the last eight weeks have been excluded from this study.

This has been clarified in the methodology section (page 3; line 111-112).

Discussion would be enriched by a deeper exploration of why MAP versus EXP domains would be differentially related to quality of life and functioning.

An explanation of why the different domains are related differently to functionality and quality of life is discussed as follows:  “it could be proposed that the two symptomatic domains of negative symptoms explore different psychopathological areas. On the one hand, EXP subscale is related to the observation of emotional expression while MAP subscale is related to more internal aspects of the emotional experience such as lack of will, lack of pleasure and absence of motivational goals that will further limit the successful interaction between people with a first psychotic episode and society” (page 7; lines 206-211).

Minor comments

Line 67 – should be “achieved”

The mistake has been fixed, thank you.

Round 2

Reviewer 1 Report

The authors have addressed all my comments appropriately, and accordingly have improved the manuscript.